# 3DOS: Towards 3D Open Set Learning – Benchmarking and Understanding Semantic Novelty Detection on Point Clouds

**Antonio Alliegro**[*]**, Francesco Cappio Borlino**[*]**, Tatiana Tommasi**

Department of Control and Computer Engineering. Politecnico di Torino, Italy
Italian Institute of Technology, Italy
`{antonio.alliegro, francesco.cappio, tatiana.tommasi}@polito.it`

## Abstract

In recent years there has been significant progress in the field of 3D learning on classification, detection and segmentation problems. The vast majority of the existing studies focus on canonical closed-set conditions, neglecting the intrinsic open nature of the real-world. This limits the abilities of robots and autonomous systems involved in safety-critical applications that require managing novel and unknown signals. In this context exploiting 3D data can be a valuable asset since it provides rich information about the geometry of perceived objects and scenes. With this paper we provide the first broad study on 3D Open Set learning. We introduce 3DOS: a novel testbed for semantic novelty detection that considers several settings with increasing difficulties in terms of semantic (category) shift, and covers both in-domain (synthetic-to-synthetic, real-to-real) and cross-domain (synthetic-to-real) scenarios. Moreover, we investigate the related 2D Open Set literature to understand if and how its recent improvements are effective on 3D data. Our extensive benchmark positions several algorithms in the same coherent picture, revealing their strengths and limitations. The results of our analysis may serve as a reliable foothold for future tailored 3D Open Set methods.

## 1  Introduction

Most existing machine learning models rely on the assumption that train and test data are drawn *i.i.d.* from the same distribution. While reasonable for lab experiments, this assumption frequently fails to hold when models are deployed in the open world, where a variety of distributional shifts with respect to the training data can emerge. For example, new object categories may induce a semantic shift, or data from new domains may give rise to a covariate shift [53, 54, 36]. Such cases can occur separately or jointly, and the test samples that differ from what was observed during training are generally indicated as out-of-distribution (OOD) data. These data may become extremely dangerous for autonomous agents as testified by the numerous accidents involving self-driving cars that misbehaved when encountering anomalous objects in the streets[2]. To avoid similar risks it is of paramount importance to build robust models capable of maintaining their discrimination ability over the closed set of known classes while rejecting unknown categories. Solving this task is challenging for existing deep models: their exceptional closed set performance hides miscalibration [16] and over-confidence issues [33]. In other words, their output score cannot be regarded as a reliable measure of prediction correctness. This drawback has been largely discussed in the 2D visual

---

[*]Equal contribution

[2]https://edition.cnn.com/2021/08/27/cars/toyota-self-driving-vehicle-paralympics-accident/index.html
https://www.foxnews.com/auto/tesla-smashes-overturned-truck-autopilot

36th Conference on Neural Information Processing Systems (NeurIPS 2022) Track on Datasets and Benchmarks.

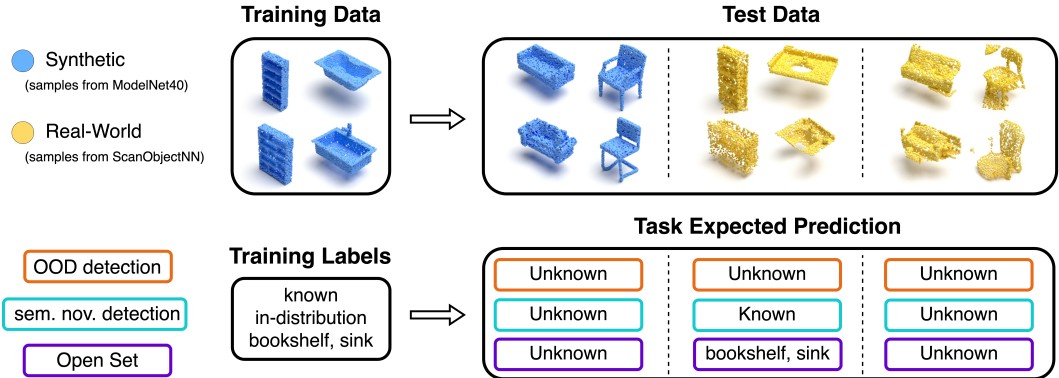

Figure 1: Schematic illustration of the *OOD detection*, *semantic novelty detection* and *Open Set* tasks on 3D data.

learning literature [41, 7, 57, 45] as its solution would enable the use of powerful deep models for many real-world tasks. In this context, and particularly for many safety-critical applications such as self-driving cars, 3D sensing is a valuable asset, providing detailed information about the geometry of sensed objects that 2D images cannot capture. However 3D literature in this field is still in its infancy, with only a small number of works which have just started to scratch the surface of the problem by focusing on particular sub-settings [31, 3]. With this work, we draw the community's attention to 3D Open Set learning, which entails developing models designed to process 3D point clouds that can recognize test samples from a set of known categories while avoiding prediction for samples from unknown classes. Our contributions are: 1) we propose 3DOS, the first benchmark for 3D Open Set learning, considering several settings with increasing levels of difficulty. It includes three main tracks: Synthetic, Real to Real, and Synthetic to Real. The first is meant to investigate the behavior of existing Open Set methods on 3D data, the other two are designed to simulate real-world deployment conditions; 2) we build a coherent picture by putting together the existing literature from OOD detection and Open Set recognition in 2D and 3D; 3) we analyze the performance of these methods to discover which is the state-of-the-art for 3D Open Set learning. We highlight their advantages and limitations and show that often a simple representation learning approach is enough to outperform sophisticated state-of-the-art methods.
Our code and data are available at `https://github.com/antoalli/3D_OS`.

## 2   Related Work

We provide an overview of existing literature on *OOD detection* and *Open Set learning*. The difference between these two tasks is often neglected, but it is important to point it out (see Fig. 1). In OOD detection it is sufficient to identify and reject samples with any distribution shift with respect to the training data. In the particular case of *semantic novelty detection*, the concept of novelty is limited to the categories not seen during training, regardless of the specific domain appearance of the observed samples. Besides separating data of known classes from those of unknown classes, Open Set recognition requires performing a class prediction over the known categories.

**Discriminative Methods.** By training a model with multi-class supervision we expect to get low uncertainty on in-distribution (ID) data and high uncertainty for OOD samples. Thus, a baseline approach may consist in using the maximum softmax probability (MSP) as a *normality score* to separate known and unknown instances [18]. However, deep models suffer from over-confidence [33] and their prediction outputs need some re-calibration to be considered as uncertainty scoring functions. ODIN [27] exploited temperature scaling and input pre-processing to better separate ID from OOD samples. In [28] the authors showed how to derive Energy scores from the prediction output, demonstrating that they are better aligned with the probability density of the inputs and are less prone to over-confidence. Instead of considering the output, GradNorm [21] focused on the network's gradients showing that their norm carries distinctive signatures to amplify the ID/OOD separability. ReAct [41] proposed to further increase this separability by rectifying the internal network activations. Finally, a very recent work has discussed how the normalized softmax probabilities can be replaced by the maximum logit scores (MLS), resulting in an approach competitive with other more complex

strategies [45]. We highlight that all the methods of this discriminative family are applied post-hoc on the closed set classifier, meaning that the original training procedure and objective are not modified. Thus, the models maintain their ability to distinguish among the known classes and are suitable for Open Set recognition.

**Density and Reconstruction Based Methods.** Density-based methods are trained to model the distribution of known data. Input samples are then identified as unknown if lying in low-likelihood regions. Several works have exploited generative models for OOD detection with novelty metrics that range from basic sample reconstruction [1, 10] to more complex likelihood ratio and regret [35, 51]. Still, generative models can be difficult to train, and their performance is frequently lower than that of discriminative ones. Recently a hybrid approach proposed to combine discriminative and probabilistic flow-based learning with promising results [57].

**Outlier Exposure.** Another line of OOD approaches exploits *outlier* data available at training time. They are used to regularize the model by applying conditions on the prediction entropy [19, 56] or running outlier mining, re-sampling and filtering [9, 26, 53].

**OOD Data Generation.** In many practical cases, it is not possible to access outlier samples at training time. Thus, unknown sample synthesis is used to prepare the model for the deployment conditions [32, 14, 7, 58]. Some recent OOD approaches have also combined real outlier mining and fake outlier generation [23].

**Representation and Distance Based Methods**. Enhancing data representation may help to better characterize known data and consequently ease the identification of unknown samples. In a reliable embedding space, OOD samples should be far away from ID classes so that the distance from stored exemplars or prototypes can be used as a scoring function. Existing approaches focus on two aspects: how to learn a good representation and how to measure distances. Self-supervised, contrastive and prototypes learning methods are of the first kind and generally rely on cosine similarity [43, 8, 40]. Other solutions build on discriminative models, but instead of considering the prediction output, they focus on the learned features and evaluate sample distances by using different metrics like $L^2$ norm, layer-wise Mahalanobis, or similarity metrics based on Gram matrices [20, 25, 38].

All the references mentioned above come from the 2D literature. Up to our knowledge 3D OOD detection and Open Set problems have been studied only by a handful of works. A VAE approach for reconstruction-based 3D OOD detection is provided in [31], together with an analysis on seven classes of the ShapeNet dataset [6], each used in turn as unknown. The study considers different VAE normality scores but does not compare with other baselines. In [3] the authors distilled knowledge from a large teacher network while also adding data produced by mixing training samples to define an unknown class. The authors focus on building a lightweight model and do not include comparisons with other Open Set methods. Moreover the classes in the used known/unknown datasets (ModelNet10/40 [49]) significantly differ in pose and headings which makes their separation trivial [24]. Two other works refer to 3D Open Set object detection and segmentation, but their objective is mainly clustering to aggregate points into object instances [5, 48].

## 3  3DOS Benchmark

Object recognition on 3D data is much more challenging than on 2D samples, with the main issues originating from the lack of color, texture, and of the general context in which the objects usually appear in images (see Fig. 2). When the goal is to evaluate whether a certain instance belongs to a known or novel class, all these cues provide crucial pieces of evidence, but without them, the task becomes really difficult. Data rescaling, resolution and noise can also significantly influence the final prediction. We dedicate our work to investigating all these aspects within the task of 3D Open Set learning: in the following we formalize the problem, introduce several testbeds and present an extensive experimental analysis.

### 3.1  Preliminaries

**Problem formulation.** We consider the labeled set $\mathcal{S} = \{x^s, y^s\}_{s=1}^N$ drawn from the training distribution $p_{\mathcal{S}}$, and we indicate as *known* all the classes $y^s \in \mathcal{Y}^s$ covered by this set. A model trained on $\mathcal{S}$ is later evaluated on the test set $\mathcal{T} = \{x^t\}_{t=1}^M$ drawn from the distribution $p_{\mathcal{T}}$. In the Open Set scenario train and test distributions differ in terms of semantic content: for the test data labels

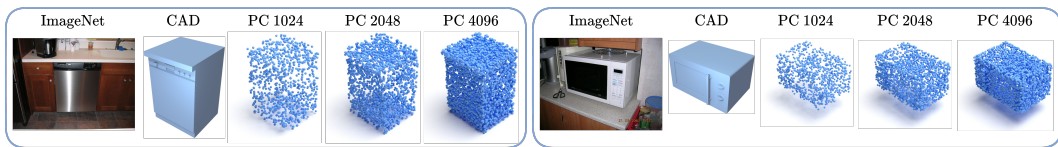

Figure 2: By looking at the point clouds of a dishwasher and microwave it might be very difficult to understand if they are the same object or not. Differently from the images, point clouds capture the object geometry, but they miss the original scale as well as color, texture, and object context which are naturally present in images.

$y^t \in \mathcal{Y}^t$ it holds $\mathcal{Y}^s \neq \mathcal{Y}^t$. More specifically, we consider a partial overlap between the two sets $\mathcal{Y}^s \subset \mathcal{Y}^t$ and the test classes which do not appear in the *known* classes set $\mathcal{Y}^s$ are therefore *unknown*. A reliable semantic novelty detection model trained on $\mathcal{S}$ should output for each test sample $x^t$ a normality score representing its probability of belonging to any of the known classes in $\mathcal{Y}^s$. An Open Set model must also provide an output probability distribution over each of the classes in $\mathcal{Y}^s$.

**Performance Metrics.** We evaluate the ability to detect unknown samples in test data by exploiting two metrics: **AUROC** and **FPR95**. Given that the detection of unknown samples is a binary task, both metrics are based on the concepts of True Positive (TP), False Positive (FP), True Negative (TN), and False Negative (FN). The AUROC (the higher the better) is the Area Under the Receiver Operating Characteristic Curve. The ROC curve is a graph showing the TP rate (TPR) and the FP rate (FPR) plotted against each other [18] when varying the normality threshold. As a result, the AUROC is a threshold-free metric, and it can be interpreted as the probability that a known test sample has a greater normality score than an unknown one. The FPR95 (the lower the better) is the FP Rate at TP Rate 95%, sometimes referred to as FPR@TPRx with x=95%. This metric is based on the choice of a normality threshold so that 95% of positive samples are predicted as positives (TPR=TP/TP+FN). Then the false positive rate (FPR=FP/FP+TN) is computed using this threshold. For Open Set methods we also evaluate their ability to correctly classify known data by computing their classification accuracy (**ACC**). Further metrics are reported in the supplementary material.

**Datasets.** We build the 3DOS Benchmark on top of three well known 3D objects datasets: ShapeNet-Core [6], ModelNet40 [49] and ScanObjectNN [44].

*ShapeNetCore* contains 51,127 meshes from synthetic instances of 55 common object categories. In our analysis we adopt ShapeNetCore v2 and use the official training (70%), validation (10%) and test (20%) splits. All objects are consistently aligned in pose. Having consistent alignment between different semantic categories is fundamental to avoid any bias that could lead to trivial inter-categories discrimination. The point clouds are obtained by uniformly sampling points from the mesh surface and normalized to fit within a unit cube centered at the origin. In our analysis we merge telephone and cellphone categories since they share similar semantic content, thus obtaining a total of 54 categories. *ModelNet40* [49] contains 12311 3D CAD models from 40 man-made object categories. We use the official dataset split, consisting of 9843 train and 2468 test shapes according to [34]. We obtain a point cloud from each CAD model by uniformly sampling points from the faces of the synthetic mesh. Each point cloud is then centered in the origin and scaled to fit within a unit cube. *ScanObjectNN* [44] contains 2902 3D scans of real-world objects from 15 categories. Specifically, we consider the original `OBJ_BG` split in which 3D scans are affected by acquisition artifacts such as vertex noise, non-uniform density, missing parts and occlusions. Data samples are already in the form of point clouds with 2048 points each and include the foreground object as well as background and other interacting objects which are absent in the synthetic instances of ModelNet and ShapeNet.

## 3.2 Benchmark Tracks

3DOS includes three main Open Set tracks. The *Synthetic Benchmark* is designed to assess the performance of existing methods in the presence of semantic shift, while the more challenging *Synth to Real Benchmark* covers both semantic and domain shift, with train and test samples that are respectively drawn from synthetic data (Modelnet40) and real-world data (ScanObjectNN). Finally, the *Real to Real Benchmark* represents an intermediate case with semantic shift among training and test data and noisy samples (from ScanObjectNN) in both sets.

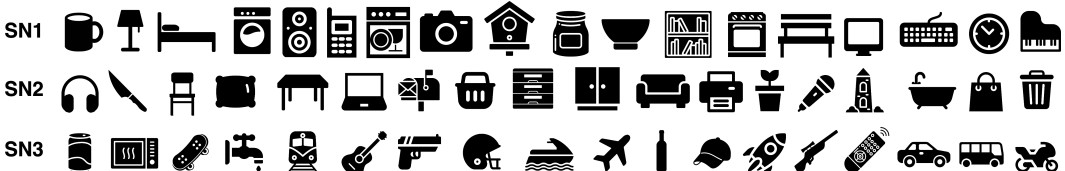

Figure 3: Visualization of the object categories in each of the sets of the Synthetic Benchmark. **SN1**: mug, lamp, bed, washer, loudspeaker, telephone, dishwasher, camera, birdhouse, jar, bowl, bookshelf, stove, bench, display, keyboard, clock, piano. **SN2**: earphone, knife, chair, pillow, table, laptop, mailbox, basket, file cabinet, sofa, printer, flowerpot, microphone, tower, bag, trash bin. **SN3**: can, microwave, skateboard, faucet, train, guitar, pistol, helmet, watercraft, airplane, bottle, cap, rocket, rifle, remote, car, bus, motorbike.

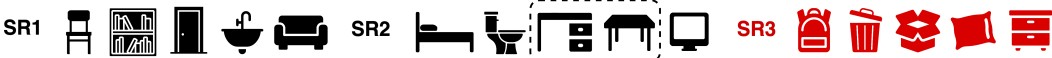

Figure 4: Visualization of the object categories in each of the sets of the Synthetic to Real Benchmark. **SR1**: chair, shelf, door, sink, sofa. **SR2**: bed, toilet, desk, table, display. **SR3**: bag, bin, box, pillow, cabinet.

**Synthetic Benchmark.** For our synthetic testbed we employ ShapeNetCore [6] dataset and we split it into 3 not overlapping (*i.e.* semantically different) category sets of 18 categories each. We dub them as SN1, SN2 and SN3 (see Fig. 3 for the list of categories belonging to each set).
We obtained three scenarios of increasing difficulty by simply selecting each of the SN-Sets in turn as *known* class set and considering the remaining two category sets as *unknown*. For this track models are trained on the train split of the known classes set and evaluated on the test split of both known and unknown classes.

**Synthetic to Real Benchmark.** To define our Synthetic to Real-World cross-domain scenario, we employ synthetic point clouds from ModelNet40 [49] for training while we test on real-world point clouds from ScanObjectNN [44]. We choose to adopt ModelNet40 (instead of ShapeNetCore) because it has a better overlap with ScanObjectNN and previous works already considered the same cross-domain scenario in the context of point cloud object classification [2, 4]. We define three different category sets: SR1, SR2, and SR3 as described in Fig. 4. The first two sets are composed by matching classes of ModelNet40 and ScanObjectNN. The third set (SR3) is instead composed by ScanObjectNN classes without such a one-to-one mapping with ModelNet40. Overall we have two scenarios with either SR1 or SR2 used as *known* and the other two considered as *unknown*. For this track, models are trained on ModelNet40 samples of the known classes set and evaluated on the ScanObjectNN samples of both known and unknown classes.

**Real to Real Benchmark.** For this last case we exploited the same SR category sets created from ScanObjectNN described above. Specifically, each of them is used as *unknown* in the test set, while the other two are divided into train and test and used as *known* classes.

### 3.3 Evaluated Methods

We consider several approaches from the families of methods described in Sec. 2.

**Discriminative Methods.** All these methods are built on top of a standard closed set classifier trained with cross-entropy. For our analysis we select the **MSP** [18] baseline, as well as its maximum logit score variant (**MLS**) [45]. We further consider **ODIN** [27], **Energy** [28], **GradNorm** [21] and **ReAct** [41].

**Density and Reconstruction Based Methods.** We select two methods from this group. We test a **VAE** model with reconstruction based scoring by following one of the few existing works on 3D anomaly detection [31]. This is the only unsupervised approach in our analysis, and thus performs only OOD detection without providing predictions over known classes. The second approach is based on Normalizing Flow (**NF**). We took inspiration by the 2D Open Set state-of-the art OpenHybrid [57] and the anomaly detection method DifferNet [37]. Specifically we train a NF model consisting of 8 coupling blocks [13] on top of the same feature embedding used by a cross-entropy classifier. The training objective consists in the maximization of the log-likelihood of training samples, and the

Table 1: Results on the Synthetic Benchmark track. Each column title indicates the chosen known class set, the other two sets serve as unknown.

| | Synthetic Benchmark - DGCNN [47] | | | | | | | | Synthetic Benchmark - PointNet++ [34] | | | | | | | |
| | SN1 (hard) | | SN2 (med) | | SN3 (easy) | | Avg | | SN1 (hard) | | SN2 (med) | | SN3 (easy) | | Avg | |
| Method | AUROC↑ | FPR95↓ | AUROC↑ | FPR95↓ | AUROC↑ | FPR95↓ | AUROC↑ | FPR95↓ | AUROC↑ | FPR95↓ | AUROC↑ | FPR95↓ | AUROC↑ | FPR95↓ | AUROC↑ | FPR95↓ |
|---|---|---|---|---|---|---|---|---|---|---|---|---|---|---|---|---|
| MSP [18] | 74.0 | 83.9 | 88.6 | 62.4 | 92.9 | 43.2 | 85.2 | 63.2 | 74.3 | 82.8 | 80.0 | 78.1 | 89.7 | 52.2 | 81.3 | 71.0 |
| MLS [45] | 75.1 | 77.7 | 91.1 | 42.6 | 92.4 | 35.2 | 86.2 | 51.8 | 72.0 | 80.8 | 83.9 | 64.1 | 89.8 | 40.5 | 81.9 | 61.8 |
| ODIN [27] | 75.4 | 76.5 | 91.1 | 42.9 | 92.5 | 34.4 | 86.3 | 51.3 | 74.2 | 79.4 | 79.4 | 71.7 | 87.8 | 41.8 | 80.5 | 64.3 |
| Energy [28] | 75.2 | 77.0 | 91.2 | 41.6 | 92.3 | 36.4 | 86.2 | 51.7 | 72.1 | 81.2 | 84.0 | 64.7 | 89.8 | 39.4 | 82.0 | 61.8 |
| GradNorm [21] | 66.2 | 88.1 | 80.9 | 64.0 | 71.6 | 77.7 | 72.9 | 76.6 | 72.1 | 81.8 | 57.7 | 88.9 | 57.8 | 79.0 | 62.6 | 83.3 |
| ReAct [41] | 76.4 | 74.6 | 92.5 | 37.9 | 96.4 | 19.3 | 88.4 | 43.9 | 73.7 | 79.4 | 89.6 | 54.4 | 95.0 | 27.2 | 86.1 | 52.9 |
| VAE [31] | 67.2 | 76.9 | 69.5 | 83.4 | 94.3 | 32.4 | 77.0 | 64.2 | - | - | - | - | - | - | - | - |
| NF | 82.0 | 74.8 | 86.1 | 53.8 | **97.4** | **11.5** | 88.5 | 46.7 | 81.5 | 72.5 | 71.1 | 78.0 | 91.0 | 49.6 | 81.2 | 66.7 |
| OE+mixup [19] | 73.7 | 78.9 | 90.4 | 44.7 | 91.4 | 46.0 | 85.2 | 56.5 | 72.7 | 78.9 | 80.3 | 68.8 | 87.3 | 62.2 | 80.1 | 69.9 |
| ARPL+CS [7] | 72.9 | 84.2 | 90.7 | 47.1 | 89.5 | 89.5 | 84.4 | 73.6 | 74.8 | 80.3 | 80.7 | 72.4 | 85.4 | 50.8 | 80.3 | 67.8 |
| Cosine proto | **84.3** | **59.1** | 88.8 | 39.7 | 86.4 | 48.0 | 86.5 | 48.9 | 80.3 | 68.3 | 88.7 | 60.8 | 91.9 | 38.0 | 86.9 | 55.7 |
| CE ($L^2$) | 80.4 | 75.5 | 90.1 | **40.9** | 96.7 | 14.4 | 89.1 | **43.6** | **83.4** | **66.8** | 89.5 | 37.7 | 92.9 | 28.1 | **88.6** | **44.2** |
| SupCon [22] | 80.3 | 75.7 | 84.6 | 73.6 | 87.9 | 44.3 | 84.3 | 64.5 | 80.9 | 75.5 | 83.5 | 68.2 | 85.1 | 45.1 | 83.2 | 62.9 |
| SubArcface [11] | 81.2 | 73.4 | **91.9** | 44.0 | 94.9 | 26.5 | **89.3** | 48.0 | 79.0 | 81.2 | 82.9 | 60.3 | 89.1 | 32.8 | 83.7 | 58.0 |

Table 2: Relationship between closed and open set performance when training a discriminative model via the addition of Label Smoothing (LS). We show the results on the hard SN1 set.

| | SN1 (hard) - Synthetic Benchmark - DGCNN [47] | | | | | | | | SN1 (hard) - Synthetic Benchmark - PointNet++ [34] | | | | | | | |
| | MSP | +LS | MLS | +LS | CE ($L^2$) | +LS | Closed set Acc | +LS | MSP | MSP+LS | MLS | MLS+LS | CE ($L^2$) | +LS | Closed set Acc | +LS |
|---|---|---|---|---|---|---|---|---|---|---|---|---|---|---|---|---|
| AUROC↑ | 74.0 | 77.4 | 75.1 | 77.5 | 80.4 | 80.2 | 85.5 | 86.0 | 74.3 | 72.7 | 72.0 | 69.6 | 83.4 | 79.1 | 85.9 | 86.3 |
| FPR95↓ | 83.9 | 73.7 | 77.7 | 71.8 | 75.5 | 66.7 | | | 82.8 | 78.6 | 80.8 | 77.6 | 66.8 | 77.4 | | |

predicted log-likelihood is later used to distinguish ID and OOD samples. Differently from the VAE this model also includes a closed set classifier and thus it is applicable to the Open Set task.

**Outlier Exposure with OOD Generated Data.** Our analysis focuses on the setting where training data does not include unknown samples, thus we assess the performance of the OE approach presented in [19] by exploiting fake OOD data produced via point cloud mixup [24] (**OE+mixup**).

**Representation and Distance Based Methods.** To evaluate the effect of a carefully learned feature embedding on the identification of novel categories, we consider the state-of-the-art 2D Open Set method **ARPL+CS** [7]. It learns reciprocal points that represent the *otherness* with respect to each known class: the distance from these points is considered proportional to the probability that a sample belongs to a certain class. Moreover, the method includes confusing samples (CS) generated in an adversarial manner to represent samples of unseen classes which are equidistant from all the reciprocal points. We also test the embedding of a cosine classifier (**Cosine proto**) as done in [15], this method learns class-prototypes by maximizing the cosine similarity between each training sample and the prototype of its class. At inference time the highest cosine similarity with a known class prototype is used as a normality score. In order to learn features representation on closed set data it is also possible to use standard losses like the supervised cross-entropy or supervised contrastive [22]. In the first case we rely on the euclidean distance between the feature of the test sample and the training samples (**CE** ($L^2$)), while in the second (**SupCon**) we use the cosine distance which better reflects the contrastive training objective. In this analysis of distance based methods we also include a seemingly unconnected technique originally proposed for face recognition. The **SubArcFace** [11] approach belongs to the family of margin-based softmax methods that aim at simultaneously achieving maximal intra-class compactness and inter-class discrepancy without the drawbacks (negative sampling and large data batches) that affect the triplet and the contrastive losses. With respect to other similar strategies [12, 46], for each known class, SubArcFace identifies multiple sub-centers, and a training sample only needs to be close to one of them rather than to a single class prototype. To get the normality score we follow the same procedure adopted for SupCon. It should be noted that the last three methods listed (CE ($L^2$), SupCon, SubArcface) require training data to be available also at test time since they compute the test sample normality score as distance to the nearest train sample.

# 4 Experiments

We perform our experimental analysis with the goal of answering a set of research questions, each discussed below in separate paragraphs respectively for the Synthetic and Synthetic to Real benchmarks. Given that 3D point clouds literature counts a large number of backbones with no dominant one, we perform all main experiments with two reliable backbones: DGCNN [47] and PointNet++ [34].

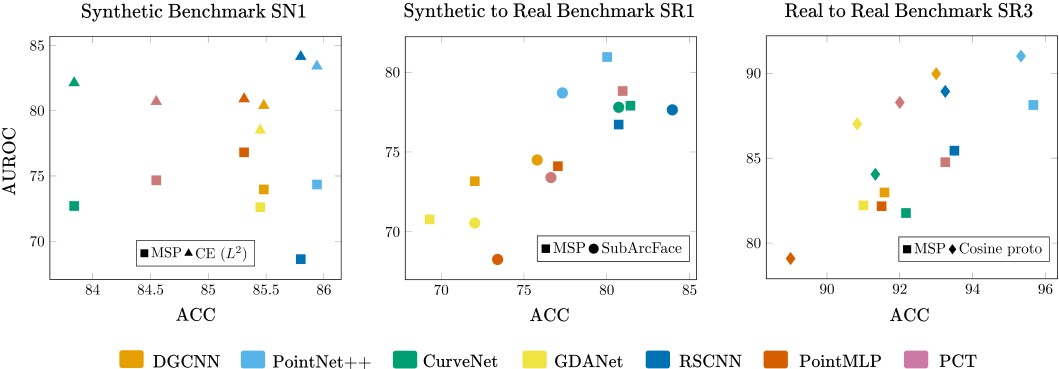

Figure 5: Correlation between AUROC and ACC performance when changing the backbone on the Synthetic Benchmark SN1 case (left), the Synthetic to Real Benchmark SR1 case (middle) and the Real to Real Benchmark SR3 case (right).

## 4.1 Implementation details

Unless otherwise specified, we use 1024 points for Synthetic point clouds (ShapeNet and ModelNet) and 2048 points for real-world point clouds (ScanObject). In the Synthetic Benchmark we augment training data with scale and translation transformations, while for the Synthetic to Real case we also augment it through random rotation around the up-axis. All models are trained with a batch size of 64 for 250 epochs, with exception of SubArcFace which is trained for 500 epochs on synthetic sets (SN1, SN2, SN3). For DGCNN experiments we use SGD optimizer with an initial learning rate of 0.1, momentum and weight decay are set respectively to 0.9 and 0.0001. With PointNet++ we employ Adam optimizer and set the initial learning rate to 0.001. Each experiment is repeated with three different seeds, we take results from the last epoch model and average across runs. Our code is implemented in PyTorch 1.9, experiments run on an HPC cluster with NVIDIA V100 GPUs. All models are trained on a single GPU except for SupCon and ARPL methods. To facilitate reproducibility we provide a complete list and discussion of the analyzed methods hyperparameters in the supplementary material.

## 4.2 Synthetic Benchmark

**How do OOD detection methods perform on 3D semantic novelty detection?** In these experiments we analyse the performance of OOD detection and Open Set methods on the Synthetic Benchmark. We report results in Tab. 1, following the same four-group organization of Sec.3.3.

*Discriminative Methods.* We consider MSP as the main baseline, but we include also its variant MLS [45]. We can see that the latter is a strong baseline, as it is often on par or better than more complex state-of-the-art methods (*e.g.* ODIN, Energy, GradNorm). In general, all methods of this group manage to improve the MSP baseline results both in terms of AUROC and FPR95, with the only exception of GradNorm. The peculiarity of this approach is that it relies on gradients extracted at test time from the network layers to compute the normality score. We hypothesize that the substantial difference between 2D (for which the model has been originally designed) and 3D network architectures may be responsible for the observed poor performance. ReAct consistently outperforms all the others with both the DGCNN and PointNet++ backbones.

*Density and Reconstruction Based Methods.* VAE results are far below the MSP baseline, this could be expected since it is the only unsupervised method in the table. It should be noticed that its encoder matches neither with DGCNN nor with PointNet++. It is composed of graph convolutional layers while the decoder is inspired to FoldingNet [55]. We still include VAE results in Tab. 1 and Tab. 3, regardless of its peculiarities, but we report its numbers with a different color. On the other hand, NF is quite sensitive to backbone choice; it performs well when trained on top of the semantically rich embedding extracted by the DGCNN, but it underperforms when trained on the local features embedding of PointNet++.

*Outlier Exposure with OOD Generated Data.* Comparing with the MSP baseline we observe that for both backbones the OE finetuning produces a slight improvement in terms of FPR95 while the AUROC does not show gains.

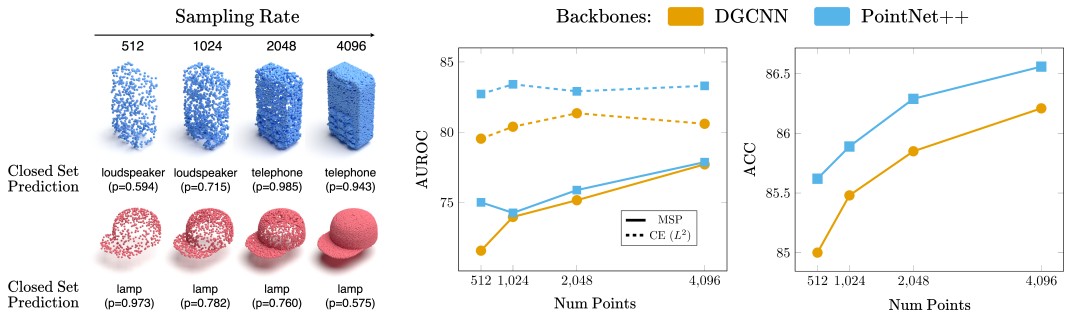

Figure 6: Analysis of the sampling rate influence on the hard SN1 set. Left: the blue telephone and red cap are known and unknown test samples for a DGCNN classifier trained on SN1. We show the class with the highest probability (p) assignment predicted by the classifier. The known object is correctly recognized as the sampling rate increases, while the confidence in the unknown object decreases, supporting rejection. Right: AUROC and ACC trends when varying the number of sampled points.

*Representation and Distance Based Methods.* The ARPL+CS [7] approach is the current state-of-the-art 2D Open Set method, however the results indicate that it does not work as well on 3D data and it is easily outperformed by the much simpler Cosine proto. Interestingly, the simple CE ($L^2$) method built on a standard cross entropy classifier obtains promising results for both backbones outperforming all the competing methods. The same considerations done for ARPL+CS hold for SupCon, which has been already successfully used in the past for OOD detection in 2D [43, 39]. We believe this is due to the fact that synthetic data are very clean and lack the variability required to build a good contrastive embedding. A better result can be obtained through SubArcFace which builds a similar feature embedding but is less computationally expensive and converges more easily. Given its state-of-the-art performance, for the following analyses we will primarily focus CE ($L^2$), along with MSP and MLS as baselines.

**What is the effect of improving closed-set classification on 3D semantic novelty detection?** A recent work has put under the spotlight the correlation between the closed accuracy of discriminative methods with their open set recognition performance on images [45]. To verify this trend on 3D data we run two sets of experiments.

A first analysis is done by exploiting a standard regularization technique as label smoothing [42]. Tab. 2 shows that LS provides a small closed set accuracy improvement for both backbones as well as some improvement also on AUROC and FPR95 when using the DGCNN backbone. It overcomes the results of ReAct (AUROC:76.4, FPR95:74.6), but remains still worse than Cosine proto which has the best performance on this set. With PointNet++ the advantage is evident only in FPR95 for MLS and MSP, but the open set performance decreases both in terms of AUROC and FPR95 for CE ($L^2$). A second evaluation is done by changing the network backbone. We experiment with a range of distinct architectures beyond the already considered DGCNN and PointNet++. Specifically, we tested CurveNet [50], GDANet [52], RSCNN [29], pointMLP [30] and PCT [17]. In particular the latter exploits Transformer blocks for point cloud learning and has recently achieved state-of-the-art performance for 3D object classification and segmentation. The results in Fig. 5 (left) show that for various 3D backbones the open set performance is not strictly linked to the closed set one. In particular, while RSCNN reaches one of the top closed set accuracy results, its MSP AUROC is the worst one.

**Is 3D semantic novelty detection affected by the point cloud density?** We investigate the impact of the point cloud sampling rate. We run experiments with different point cloud sizes: 512, 1024, 2048, and 4096. For each experiment we fix the number of points (*e.g.* 512) at both training and evaluation. Point clouds with higher sampling rates are more detailed and fine-grained structures become visible at the cost of higher computational complexity. We show in Fig. 6 (left) some visualizations of the influence of the sampling on the visibility of local details, which are important for both known vs unknown discrimination and closed set classification. In the right part Fig. 6, we report the results of closed and open set performance for MSP and CE ($L^2$). While the closed set accuracy grows as the number of sampled points increases, there is no corresponding increase for the open set performance of CE ($L^2$).

Table 3: Results on the Synthetic to Real Benchmark track. Each column title indicates the chosen known class set, the other two sets serve as unknown.

| | Synth to Real Benchmark - DGCNN [47] | | | | | | Synth to Real Benchmark - PointNet++ [34] | | | | | |
| | SR 1 (easy) | | SR 2 (hard) | | Avg | | SR 1 (easy) | | SR 2 (hard) | | Avg | |
| *Method* | AUROC↑ | FPR95↓ | AUROC↑ | FPR95↓ | AUROC↑ | FPR95↓ | AUROC↑ | FPR95↓ | AUROC↑ | FPR95↓ | AUROC↑ | FPR95↓ |
|---|---|---|---|---|---|---|---|---|---|---|---|---|
| MSP [18] | 72.2 | 91.0 | 61.2 | 90.3 | 66.7 | 90.6 | 81.0 | 79.6 | 70.3 | 86.7 | 75.6 | 83.2 |
| MLS | 69.0 | 92.2 | 62.4 | 88.9 | 65.7 | 90.5 | 82.1 | 76.6 | 67.6 | 86.8 | 74.8 | 81.7 |
| ODIN [27] | 69.0 | 92.2 | 62.4 | 89.0 | 65.7 | 90.6 | 81.7 | 77.3 | 70.2 | 84.4 | 76.0 | 80.8 |
| Energy [28] | 68.8 | 92.7 | 62.4 | 88.9 | 65.6 | 90.8 | 81.9 | 77.5 | 67.7 | 87.3 | 74.8 | 82.4 |
| GradNorm [21] | 67.0 | 93.5 | 59.8 | 89.4 | 63.4 | 91.5 | 77.6 | 80.1 | 68.4 | 86.3 | 73.0 | 83.2 |
| ReAct [41] | 68.4 | 92.1 | 62.8 | 88.8 | 65.6 | 90.5 | 81.7 | 75.6 | 67.6 | 87.2 | 74.6 | 81.4 |
| VAE [31] | 68.6 | 77.0 | 57.9 | 92.3 | 63.3 | 84.6 | - | - | - | - | - | - |
| NF | 72.5 | **81.6** | 70.2 | **83.0** | 71.3 | **82.3** | 78.0 | 84.4 | 74.7 | 84.2 | 76.4 | 84.3 |
| OE+mixup [19] | 71.1 | 89.6 | 59.5 | 92.0 | 65.3 | 90.8 | 71.2 | 89.7 | 60.3 | 93.5 | 65.7 | 91.6 |
| ARPL+CS [7] | 71.5 | 90.2 | 62.8 | 89.5 | 67.1 | 89.8 | **82.8** | 74.9 | 68.0 | 89.3 | 75.4 | 82.1 |
| Cosine proto | 58.6 | 90.6 | 57.3 | 91.3 | 57.9 | 91.0 | 79.9 | **74.5** | **76.5** | **77.8** | **78.2** | **76.1** |
| CE ($L^2$) | 67.5 | 87.4 | 64.6 | 91.0 | 66.1 | 89.2 | 79.7 | 84.5 | 75.7 | 80.2 | 77.7 | 82.3 |
| SubArcFace [11] | **74.5** | 86.7 | 68.7 | 86.6 | **71.6** | 86.7 | 78.7 | 84.3 | 75.1 | 83.4 | 76.9 | 83.8 |

Table 4: Results on the Real to Real Benchmark track. Each column title indicates the chosen unknown class set, the other two sets serve as known.

| | Real to Real Benchmark - DGCNN [47] | | | | | | | | Real to Real Benchmark - PointNet++ [34] | | | | | | | |
| | SR3 (easy) | | SR2 (med) | | SR1 (hard) | | Avg | | SR3 (easy) | | SR2 (med) | | SR1 (hard) | | Avg | |
| *Method* | AUROC↑ | FPR95↓ | AUROC↑ | FPR95↓ | AUROC↑ | FPR95↓ | AUROC↑ | FPR95↓ | AUROC↑ | FPR95↓ | AUROC↑ | FPR95↓ | AUROC↑ | FPR95↓ | AUROC↑ | FPR95↓ |
|---|---|---|---|---|---|---|---|---|---|---|---|---|---|---|---|---|
| MSP [18] | 83.0 | 69.4 | 72.0 | 88.7 | 57.5 | 90.3 | 70.8 | 82.8 | 88.1 | 67.3 | 80.6 | 84.0 | 73.7 | 80.3 | 80.8 | 77.2 |
| MLS [45] | 84.9 | 58.2 | 79.0 | 81.0 | 54.0 | 92.8 | 72.6 | 77.3 | 89.4 | 53.8 | **83.4** | 73.1 | 76.4 | **75.3** | 83.0 | 67.4 |
| ODIN [27] | 84.9 | 58.2 | 79.0 | 80.9 | 54.0 | 92.8 | 72.6 | 77.3 | 90.2 | 47.9 | 83.3 | 71.7 | 76.3 | 76.8 | 83.3 | 65.5 |
| Energy [28] | 84.8 | 59.7 | 79.1 | 81.4 | 53.8 | 93.2 | 72.6 | 78.1 | 89.5 | 50.6 | 81.6 | 75.8 | 76.6 | 75.5 | 82.6 | 67.3 |
| GradNorm [21] | 77.5 | 73.3 | 73.3 | 87.4 | 51.0 | 92.9 | 67.2 | 84.5 | 88.5 | 50.7 | 77.4 | 75.3 | 75.2 | 76.8 | 80.4 | 67.6 |
| ReAct [41] | 87.6 | 54.0 | 79.0 | 78.6 | 58.9 | 93.1 | 75.1 | 75.3 | 90.3 | 48.9 | 82.4 | 75.8 | 75.4 | 77.6 | 82.7 | 67.4 |
| NF | 76.9 | 77.3 | 71.7 | 82.7 | 61.8 | 86.2 | 70.2 | 82.1 | 88.0 | 47.7 | 80.6 | **68.2** | 75.6 | 81.4 | 81.4 | 65.8 |
| OE+mixup [19] | 76.8 | 77.8 | 74.9 | 87.2 | 57.6 | 89.9 | 69.8 | 85.0 | 72.6 | 83.5 | 72.0 | 88.5 | 62.5 | 87.8 | 69.0 | 86.6 |
| Cosine proto | **90.0** | **43.7** | **78.5** | **75.3** | 65.5 | 85.7 | **78.0** | **68.2** | **91.0** | **41.0** | 82.1 | 78.2 | **77.6** | 75.6 | **83.6** | **64.9** |
| CE ($L^2$) | 83.1 | 59.3 | 74.5 | 77.2 | **67.1** | 86.8 | 74.9 | 74.4 | 85.1 | 64.4 | 78.9 | 83.9 | 73.2 | 79.1 | 79.1 | 75.8 |
| SubArcface [11] | 86.7 | 58.5 | 78.4 | 76.1 | 65.0 | **84.0** | 76.7 | 72.9 | 87.1 | 61.3 | 78.9 | 76.9 | 73.7 | 81.4 | 79.9 | 73.2 |

## 4.3 Synthetic to Real Benchmark

Training on synthetic data is fundamental, especially when only a few real-world samples are available for a given task. This is often the case for 3D point cloud learning, for which ScanObjectNN [44] is one of the largest publicly available real-world object datasets despite counting less than 3k samples. We thus analyze how models trained on synthetic data perform when tested on real-world data.

**How does the OOD detection performance trend changes when testing on Real-World data?**
Table 3 provides an overview of the results on the Synthetic to Real benchmark. W.r.t. Table 1 we notice a general degradation in performance due to the domain shift between train (synthetic) and test (real-world). Interestingly, the PointNet++ backbone seems to more robust to the domain shift than DGCNN. For example, the MSP baseline with PointNet++ outperforms the DGCNN counterpart by 8.9 pp in terms of AUROC and 7.4 in terms of FPR95. We include a more comprehensive analysis of the impact of the backbone used in the Synthetic to Real benchmark in Fig. 5 (middle). Most of the methods that performed well in the Synthetic benchmark turn out to be less robust than the simple MSP baseline, and thus can no longer outperform it. This is true for the vast majority of discriminative methods. For VAE it holds a similar discussion to what was done for the synthetic counterpart. NF performs consistently on both backbones with a clear AUROC improvement of 4.6 pp over MSP when applied on top of DGCNN and only a slight improvement for PointNet++. In the case of OE+mixup, the generated outliers used to finetune the classifier model are not representative of the real-world test domain and thus do not allow for improvement over the MSP baseline.

The results of distance based methods are highly dependent on the specific backbone chosen. Cosine proto performs particularly well on PN2, but fails miserably on DGCNN, most likely because DGCNN prototypes trained on synthetic are not representative of the real-world test distribution. A similar consideration can be done for CE ($L^2$), confirming the robustness of PointNet++ to the domain shift. Finally, SubArcFace demonstrates its reliability, as it achieves good results on both backbones and the best overall results on average.

For both MSP and SubArcFace we studied the impact of several backbones also considering the closed set performance as shown in the middle part of Fig. 5. The plot shows a linearly growing

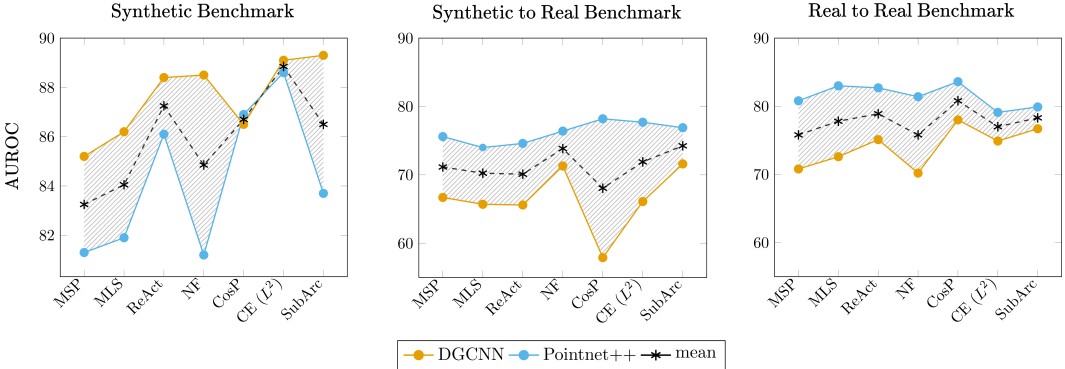

Figure 7: AUROC scores across methods and backbones for the 3DOS benchmark tracks indicated by the respective titles.

trend for both methods and the results also confirm the advantage of PointNet++ over more complex networks.

### 4.4 Real to Real Benchmark

To complete our analysis we ran the most relevant methods on the Real to Real Benchmark and present the results in Table 4. Overall, the trend for all approaches is consistent with what was observed in the Synthetic to Real case. The Cosine proto approach, which already demonstrated effectiveness with PointNet++ in the Synthetic to Real benchmark, now ranks first for both DGCNN and PointNet++. We also highlight that PointNet++ maintains better performance than DGCNN confirming its robustness when dealing with noisy and corrupted real-world data.

For both MSP and Cosine proto we studied the impact of several backbones also considering the closed set performance as shown in the right part of Fig. 5.

## 5 Conclusions

We presented 3DOS, the first benchmark for 3D Open Set learning that takes into account several settings and three scenarios with different types of distributional shifts. Our analysis reveals that cutting-edge 2D Open Set methods do not easily transfer their state-of-the-art performance to 3D data, with simple representation learning approaches such as CE $(L^2)$, SubArcFace and Cosine proto often outperforming them. Furthermore, the performances of the 3D Open Set methods depend on the chosen backbone: PointNet++ has proven to be extremely robust in processing real-world data, even across domains, outperforming more recent and complex networks. The point density may be an issue for baseline approaches but has a minimal impact on distance-based strategies as CE $(L^2)$. Finally, Open Set on 3D data becomes extremely difficult when dealing with the combination of semantic and domain shift.

Figure 7 depicts a summary overview of the three studied scenarios indicating how the Synthetic to Real is the most challenging case, followed by the Real to Real and finally by the Synthetic Benchmark. This confirms that the domain shift between synthetic and real data adds extra challenges over the semantic shift. Moreover, it is interesting to notice that the improvement provided by the best Open Set methods over the MLS/MSP baselines is quite visible in the Synthetic Benchmark (DGCNN SubArcFace > MLS, +3.1 AUROC), but is reduced in the Synthetic to Real (PointNet++ Cosine Proto > MSP, + 2.6 AUROC) and Real to Real cases (PointNet++ Cosine Proto > MLP, + 0.6 AUROC), which clearly asks for new approaches and reveals room for improvements.

We hope that this benchmark will serve as a solid foundation for future research in this area, pushing for the development of Open Set methods tailored for 3D data and able to exploit their peculiarity.

**Acknowledgements** We acknowledge the CINECA award under the ISCRA initiative, for the availability of high performance computing resources and support. We also acknowledge the support of the European H2020 Elise project (`www.elise-ai.eu`).

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
