# 3DOS: Towards Open Set 3D Learning – Benchmarking and Understanding Semantic Novelty Detection on Point Clouds Supplementary material

**Antonio Alliegro**\*, **Francesco Cappio Borlino**\*, **Tatiana Tommasi**
Department of Control and Computer Engineering. Politecnico di Torino, Italy
Italian Institute of Technology, Italy
`{antonio.alliegro, francesco.cappio, tatiana.tommasi}@polito.it`

## 1 Baselines details, implementation and reproducibility

We publicly release our code and data at `https://github.com/antoalli/3D_OS`. The repository also contains instructions on how to replicate all the experiments.

In the main paper we include a high-level description of all methods and the most relevant implementation details. Here we extend the description, discussing implementation choices and hyperparameters for each.

**Discriminative Models**

All the approaches in this group (MSP, MLS, ODIN, Energy, GradNorm, ReAct) share exactly the same basic cross-entropy classifier trained on known data. Section 4.1 of the main paper already specified the cardinality (number of points) of the point clouds in train and test, as well as the number of epochs, initial learning rate and learning rate scheduling policy.

*MSP & MLS* only differ for the way in which the logits of the classifier on each test sample are used to compute the normality score. MLS directly employs the maximum logit, while for MSP the logits go through a softmax function before selecting the maximum of the obtained class probabilities as the score.

*ODIN* internally exploits input perturbation and temperature scaling since both have an effect on the distribution of the softmax scores, better separating data from known and unknown classes. The most important hyperparameter here is the temperature value which we set to $T = 1000$ following the original paper's instructions [27]. The input perturbation magnitude $\varepsilon$ should be optimized through a validation set of OOD samples, still keeping its value very small to avoid detrimental effects. Considering that we do not have access to OOD data at training time we preferred to stay on the safe side, setting $\varepsilon = 0$ in all of our experiments, effectively disabling the input perturbation.

*Energy* The energy-based normality score is computed by postprocessing the network output logits and it's hyperparameter free.

*GradNorm* In order to compute a normality score, the KL divergence between the network output and a uniform distribution is backpropagated to obtain network gradients and then extract their norm. As suggested in the original paper [21], we exploit the norm of the gradients of the last layer only. No further hyperparameters are involved in this process.

*ReAct* rectifies the test-time activations of the network trained for classification and then can exploit any normality score computation strategy. By following [40] we use ReAct in combination with

---

\*Equal contribution

36th Conference on Neural Information Processing Systems (NeurIPS 2022) Track on Datasets and Benchmarks.

Energy normality score. The only hyperparameter involved is the rectification threshold value. We chose it by exploiting the known class validation samples to preserve 90% of ID activations. For the Synthetic Benchmark the known classes validation samples come from the original ShapeNetCore [6] validation split. For the Synthetic to Real Benchmark, we adopt ModelNet40 [48] known classes test set as validation, we underline that these samples are not involved in the testing phase since both Closed Set accuracy and Open Set performance are computed on ScanObjectNN [43].

**Density and Reconstruction Based Models**

*VAE* For our experiments we use the original code publicly released by the authors[2] [31], as well as their same choice on point cloud cardinality (2048 points) and hyperparameters. The encoder is composed of graph-convolutional layers: it takes as input a point cloud and outputs two 512-dimensional vectors representing the mean and variance. The decoder is a FoldingNet: it takes in input a sampled vector z from the encoded mean and variance and outputs an intermediate and a final point cloud reconstruction with respectively 1024 and 2048 points. The normality score is computed as the Chamfer Distance between the original test sample and its final reconstruction.

*NF* For this method we got inspired by [55]. The overall architecture consists of three modules: a feature encoder, a classification head, and a Normalizing Flow (NF) head. The feature encoder and classification head work together to optimize a standard cross-entropy loss. The NF head works independently on top of the feature encoder representation and it is composed of eight Real-NVP [13] coupling blocks which are trained to maximize the log-likelihood of the observed training features. At inference time we use the test sample log-likelihood as a normality score. For training NF we use the Adam optimizer with a learning rate of 0.0002 and weight decay set to 0.00001.

**Outlier Exposure with OOD Generated Data** The outlier exposure strategy described in [19] consists in training a Discriminative Model on ID training data (known) through standard cross-entropy loss, while exploiting additional OOD training data (unknown) to improve ID-OOD separability. Specifically, we start from the same cross-entropy classifier trained on known classes employed for the first group of strategies (i.e. Discriminative Models) and finetune it by minimizing the following loss function: $\mathcal{L}_{ft} = \mathcal{L}_{CE,known} + \lambda\mathcal{L}_{OE,unknown}$.

The finetuning involves a continued optimization of the cross-entropy loss on known training data $\mathcal{L}_{CE,known}$ and an outlier exposure objective $\mathcal{L}_{OE,unknown}$ on unknown training data. The goal of the outlier exposure objective is to minimize the KL divergence between the Uniform and Cross-entropy distributions for unknown samples. The hyperparameter $\lambda$ controls the importance of the OE auxiliary objective and is set to $0.5$ according to the original paper [19]. The finetuning is performed for additional 100 epochs, with a learning rate reduced by a factor of 100.

For OE finetuning, OOD training data are obtained through Rigid Subset Mix (RSMix) [24] of known class samples, some examples of the produced OOD data for the synthetic SN1 set are shown in Fig. 2.

**Representation and distance-based methods**

*ARPL+CS* The training process of this method involves learning a GAN model designed to generate confusing samples (CS), as well as optimizing the reciprocal points learning objective. The model needs a number of hyperparameters to keep all the learning components well-balanced and we used the same values adopted by the authors [7].

*Cosine proto* We train a simple cosine classifier by following CosFace [45] strategy and setting the imposed margin to 0. At inference time output logits correspond to the cosine similarities between the test sample and the class prototypes. The largest value is used as the normality score without introducing additional hyperparameters.

*CE ($L^2$)* With this method we aim at studying the reliability to OOD detection of the feature representation learned by the same classifier trained for the Discriminative Models. We use the inverse of the distance from the nearest training sample as each test sample normality score, without introducing additional hyperparameters.

*SupCon* requires long training with a large batch size to reach convergence [22]. With respect to the models trained for classification we double the batch size and halve the learning rate. To deal with large batches we perform distributed training on multiple GPUs and adopt Synchronized Batch

---

[2]`https://github.com/llien30/point_cloud_anomaly_detection`

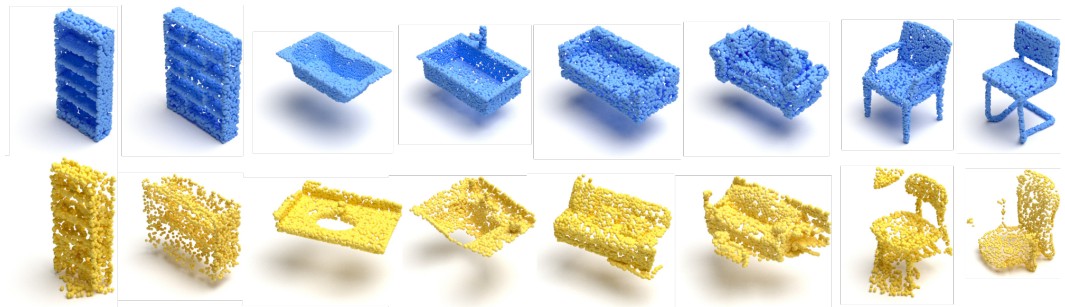

Figure 1: Qualitative visualizations: point clouds from bookshelf, sink, sofa and chair categories. Blue are synthetic point clouds from ModelNet40, yellow are real-world from ScanObjectNN

Normalization. We also increase the number of epochs to 2000, using a linear warmup in the first 100. During deployment the normality score is the cosine similarity of each test sample to its nearest training sample. The SupCon learning objective builds a hyperspherical feature space in which class clusters are compact and well separated. Similar results can be obtained through SubArcFace which exploits a much more easily optimized classification-like loss. On this basis, and also considering the poor results of SupCon on the Synthetic Benchmark, we decided to discard it in the Synthetic to Real and Real to Real Benchmarks.

*SubArcFace* learning objective seeks to maximize the cosine similarity between each training sample and one of the $K$ centers associated with its respective class, while also imposing a certain margin $m$ between different classes. We use $K = 3$ as done by the authors in the original paper [11] and set the margin to $m = 0.5$, as done in ArcFace [12] from which this hyperparameter is inherited. The normality score is computed as done for SupCon.

## 2 Synthetic to Real Benchmark: Additional Analyses

The goal of the Synthetic to Real Benchmark track is to simulate realistic deployment conditions and analyze the behaviour of Open Set methods in this context. Indeed, due to the high cost of 3D data acquisition and labelling, large synthetic datasets are commonly used to train deep neural network models which are then employed in real-world applications such as autonomous driving, augmented reality or robotics.

This strategy, although effective in lowering data collection costs, inevitably causes a covariate distribution (visual domain) shift between training and test data. As a result test samples belonging to unknown classes show both a semantic and a domain shift, while test samples belonging to known classes only show a domain shift. The necessity to distinguish between these two cases raises the difficulty of the unknown detection task. To get an idea of the difference between the synthetic and the real domains we render some point clouds in Figure 1, respectively from the known classes in the train and test of our Synthetic to Real Benchmark track.

It is evident that real-world samples (in yellow) are much noisier than synthetic ones (in blue). Moreover, real-world point clouds have background (first chair), are affected by occlusion, partiality (second chair) and interaction with other objects nearby (second sofa and first chair).

This additional covariate shift, which is present only in the Synthetic to Real Benchmark, is what makes this track the most difficult among the ones we analyzed, as highlighted in the Conclusion of the main paper.

### 2.1 AUPR metric

In the results reported in the main paper we exploited two of the most common OOD detection metrics to evaluate the unknown detection ability of the analyzed methods: AUROC and FPR95. Different metrics may be chosen for the same purpose, one of the most used being the Area Under the Precision Recall curve (**AUPR**). Similarly to AUROC this is a threshold-free metric: the *precision* $= TP/(TP+FP)$, is plotted as a function of *recall* $= TP/(TP+FN)$, for different threshold settings

Table 1: AUROC, FPR95, AUPR results on the Synthetic to Real Benchmark with PointNet++ [34]

| | Synthetic to Real Benchmark - PointNet++ [34] | | | | | | | | |
| | SR 1 | | | SR 2 | | | Avg | | |
| *Method* | AUROC ↑ | FPR95 ↓ | AUPR ↑ | AUROC ↑ | FPR95 ↓ | AUPR ↑ | AUROC ↑ | FPR95 ↓ | AUPR ↑ |
|---|---|---|---|---|---|---|---|---|---|
| MSP [18] | 81.0 | 79.6 | 79.9 | 70.3 | 86.7 | 83.9 | 75.6 | 83.2 | 81.9 |
| MLS [44] | 82.1 | 76.6 | 82.0 | 67.6 | 86.8 | 83.1 | 74.8 | 81.7 | 82.5 |
| ODIN [27] | 81.7 | 77.3 | 81.5 | 70.2 | 84.4 | 84.4 | 76.0 | 80.8 | 82.9 |
| Energy [28] | 81.9 | 77.5 | 82.0 | 67.7 | 87.3 | 83.0 | 74.8 | 82.4 | 82.5 |
| GradNorm [21] | 77.6 | 80.1 | 78.8 | 68.4 | 86.3 | 83.5 | 73.0 | 83.2 | 81.2 |
| ReAct [40] | 81.7 | 75.6 | 81.9 | 67.6 | 87.2 | 83.1 | 74.6 | 81.4 | 82.5 |
| NF | 78.0 | 84.4 | 77.0 | 74.7 | 84.2 | 86.3 | 76.4 | 84.3 | 81.7 |
| OE+mixup [19] | 71.2 | 89.7 | 70.9 | 60.3 | 93.5 | 77.9 | 65.7 | 91.6 | 74.4 |
| ARPL+CS [7] | **82.8** | 74.9 | **82.7** | 68.0 | 89.3 | 83.4 | 75.4 | 82.1 | 83.0 |
| Cosine proto | 79.9 | **74.5** | 81.2 | **76.5** | **77.8** | **88.1** | **78.2** | **76.1** | **84.7** |
| CE ($L^2$) | 79.7 | 84.5 | 78.4 | 75.7 | 80.2 | 87.3 | 77.7 | 82.3 | 82.9 |
| SubArcFace [11] | 78.7 | 84.3 | 77.2 | 75.1 | 83.4 | 86.1 | 76.9 | 83.8 | 81.6 |

Table 2: Synthetic to Real Benchmark with real-world augmentations

| | Synthetic to Real Benchmark - DGCNN [46] | | | | | | Synthetic to Real Benchmark - PointNet++ [34] | | | | | |
| | SR 1 | | SR 2 | | Avg | | SR 1 | | SR 2 | | Avg | |
| *Method* | AUROC↑ | FPR95↓ | AUROC↑ | FPR95↓ | AUROC↑ | FPR95↓ | AUROC↑ | FPR95↓ | AUROC↑ | FPR95↓ | AUROC↑ | FPR95↓ |
|---|---|---|---|---|---|---|---|---|---|---|---|---|
| MSP [18] | 72.2 | 91.0 | 61.2 | 90.3 | 66.7 | 90.6 | 81.0 | 79.6 | 70.3 | 86.7 | 75.6 | **83.2** |
| MSP (+RW Augm) | 82.1 | 76.0 | 65.8 | 92.8 | 73.9 | 84.4 | 76.5 | 81.8 | 74.6 | 85.9 | 75.5 | 83.9 |
| SubArcFace [11] | 74.5 | 86.7 | 68.7 | 86.6 | 71.6 | 86.7 | 78.7 | 84.3 | 75.1 | 83.4 | **76.9** | 83.8 |
| SubArcFace (+RW Augm) | 81.3 | 77.4 | 68.8 | 84.7 | **75.1** | **81.1** | 76.9 | 83.7 | 73.0 | 89.5 | 75.0 | 86.6 |

and then the area under the resulting curve is computed, with a high value (near to 1) indicating a good known-unknown separation and a low one (near to 0) highlighting bad performance. Differently from the AUROC, the AUPR takes care of the possible unbalance between positive and negative classes by adjusting for the base rates. Specifically, we computed the AUPR considering the unknown samples as positive: given its complementary polarity with AUROC we expect it to provide further information.

We report the results of the best performing backbone on the most difficult benchmark track, i.e. we run with PointNet++ [34] on the Synthetic to Real Benchmark: see Table 1. According to AUPR, the top performing methods are the same ones that AUROC and FPR95 highlighted as best.

## 2.2 OE+mixup in the Synthetic to Real Benchmark

Figure 2 presents point cloud instances obtained via mixup. This strategy is used to create data that can be exploited as OOD during training via outlier exposure. However, as it can be noticed, shape mixing inevitably introduces some artefacts that resemble noise, missing parts and background, typical of real-world data. We believe that in the synthetic-to-real experiments this may introduce some confusion rather than helping in separating known and unknown classes, as it pushes the model to believe that all corrupted samples belong to unknown classes. This reflects in the poor *OE+mixup* results reported in Tab. 3 of the main paper.

## 2.3 Corruption-based data augmentation

The results of the Synthetic to Real benchmark highlight the impact of the domain shift on the open set performance. Indeed, synthetic point clouds exhibit a clean geometry and have no background. Differently, real-world point clouds are affected by partiality, and occlusion, cluttered with noise and background points. A possible solution to partially bridge this domain shift consists in trying to emulate these kinds of corruptions at training time via tailored data augmentation functions.

We experiment with this solution by adopting the *Occlusion* and *LIDAR* augmentations from [57]. Fig. 3 shows some examples of the results obtained using these transformations. Comparing them with point clouds in the second row of Fig. 1 it is possible to notice some differences. Considering both DGCNN and PointNet++ backbones we perform experiments with such augmented training

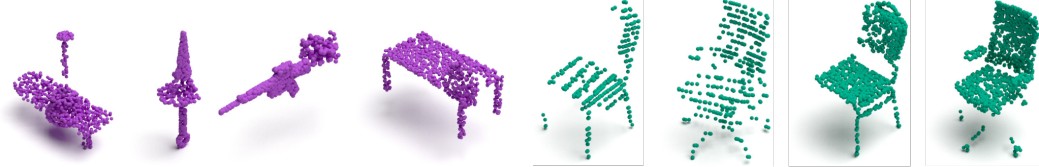

Figure 2: Examples of RSMix [24] between known samples of the synthetic SN1 set. We employ these mixed point clouds as training OOD data in OE experiments

Figure 3: Examples of ModelNet point clouds augmented with LIDAR (first two) and Occlusion (last two) corruptions from [57]

Table 3: Results on the Synthetic to Real Benchmark track when varying the number of test points. Reported results are average over the two possible scenarios (SR1, SR2).

| | Synth (1024) to Real (2048) - Avg | | | | Synth (1024) to Real (1024) - Avg | | | | Synth (1024) to Real (512) - Avg | | | |
| | DGCNN | | PointNet++ | | DGCNN | | PointNet++ | | DGCNN | | PointNet++ | |
| Method | AUROC↑ | FPR95↓ | AUROC↑ | FPR95↓ | AUROC↑ | FPR95↓ | AUROC↑ | FPR95↓ | AUROC↑ | FPR95↓ | AUROC↑ | FPR95↓ |
|---|---|---|---|---|---|---|---|---|---|---|---|---|
| MSP [18] | 66.7 | 90.6 | 75.6 | 83.2 | 70.2 | 86.7 | 76.5 | 84.0 | 58.5 | 92.1 | 74.3 | 84.4 |
| MLS [44] | 65.7 | 90.5 | 74.8 | 81.7 | 70.4 | 86.4 | 76.4 | 79.0 | 61.9 | 89.7 | 75.2 | 80.9 |
| ODIN [27] | 65.7 | 90.6 | 76.0 | 80.8 | 70.1 | 87.4 | 78.3 | 81.0 | 57.4 | 93.3 | 76.1 | 81.3 |
| Energy [28] | 65.6 | 90.8 | 74.8 | 82.4 | 70.6 | 86.9 | 77.6 | 78.9 | 62.5 | 88.7 | 76.3 | 80.4 |
| GradNorm [21] | 63.4 | 91.5 | 73.0 | 83.2 | 70.5 | 85.8 | 76.5 | 78.0 | 62.2 | 89.8 | 75.5 | 79.8 |
| ReAct [40] | 65.6 | 90.5 | 74.6 | 81.4 | 67.5 | 88.3 | 74.3 | 80.9 | 60.8 | 90.4 | 73.1 | 82.8 |
| Cosine proto | 57.9 | 90.9 | **78.2** | **76.1** | 70.0 | 86.3 | 76.2 | 77.7 | 61.1 | 88.8 | 74.9 | 79.5 |
| CE ($L^2$) | 66.0 | 89.2 | 77.7 | 82.3 | **73.9** | **82.2** | 77.0 | 84.2 | **64.9** | **88.4** | 72.4 | 86.5 |
| SubArcface [11] | **71.6** | **86.7** | 76.9 | 83.8 | 61.9 | 88.6 | **78.5** | **76.4** | 55.0 | 93.1 | **76.8** | **76.6** |

data for both the simple MSP baseline and SubArcFace method. Results for these experiments are presented in Tab. 2 where we refer to the models trained with the augmented training set as (+RW Augm). DGCNN highly benefit from the real-world tailored data augmentation, obtaining an AUROC improvement of +7.2pp and +3.5pp respectively for MSP and SubArcFace methods. PointNet++, on the other hand, has already proven its robustness in synthetic to real-world scenario and does not benefit from the tailored augmentation schema.

## 2.4 Number of points at test time

The visual domain shift between training and test conditions that appear in the Synthetic to Real case may include also a difference in the cardinality of points. In our Synthetic to Real Benchmark we followed the same procedure adopted for the Synthetic Benchmark case: we use 1024-dim synthetic points clouds during training, with points randomly sampled from the surface of the ModelNet40 meshes. The trained model is then evaluated solely on real-world samples from the ScanObject dataset. Real-world data samples come directly in the form of 2048-dim point clouds. For each sample, out of the 2048 points, we do not know a priori which are belonging to the foreground object, background, or other interacting objects. In order to ease results replication, we originally decided to avoid random subsampling and used at test time 2048-dim real-world points clouds in all our Synthetic to Real experiments. We are now interested in analyzing how the number of points used during inference influences the results and thus we test with 1024 and 512 and report results in Tab. 3. Looking at the results we can conclude that forcing train and test data to have the same number of points (1024-1024) slightly reduces the domain shift and provides a small performance improvement with respect to the remaining cases that present an asymmetry in the point cloud cardinality (1024-2048, 1024-512).

## 3 Analysis of the error margin

All the experimental results presented in the main paper are average over three experiments repetitions with different seeds. In Fig. 4 we report both the average and standard deviation for the MSP baseline and the methods that presented top results, respectively CE ($L^2$) for the *Synthetic*, SubArcFace for the *Synthetic to Real* and Cosine proto for the *Real to Real* Benchmarks. By looking at the error bars, we can see that in the Synthetic Benchmark the standard deviation is quite small and the advantage of CE

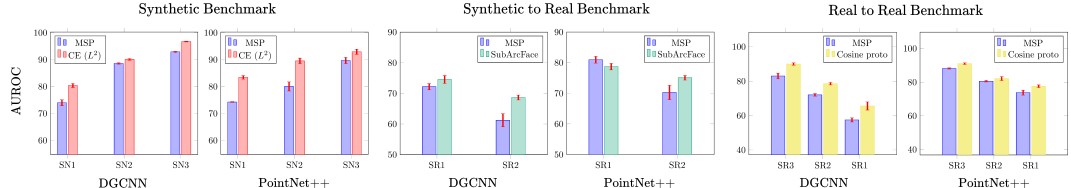

Figure 4: Error margin analysis. State-of-the-art and baseline AUROC results on the three 3DOS tracks: Synthetic, Synthetic to Real and Real to Real Benchmarks. Red error bars represent the standard deviation around the mean value.

($L^2$) over MSP is always statistically significant. In the Synthetic to Real Benchmark the performance gap is lower, especially for the SR1 case, where MSP outperforms SubArcFace with the PointNet++ backbone. With both backbones however the results of the baseline and the state-of-the-art approach are within the error margin. SubArcface however has significantly better performance on the SR2 setting with also a smaller standard deviation.

When inspecting the Real to Real Benchmark results the performance gap between the baseline and the top-performing method is most noticeable with DGCNN. Clearly, the improved performance obtained by using a more robust backbone such as PointNet++ reduces the importance of selecting a stronger learning approach. In any case, the error margin is quite low and appears to decrease as performance improves, highlighting the high reliability of the results obtained in this Benchmark.

# 4 Further discussion

## 4.1 Limitations

Some limitations of our work are directly inherited from the 3D computer vision field. The benchmark would undoubtedly benefit from the inclusion of a large-scale real-world dataset; however, such a dataset would have to be purposefully collected and curated because it is currently unavailable. In the last years, huge progress has been made in 3D deep learning literature. However, most of the recent works exclusively focus on synthetic scenarios, now exhibiting a trend of performance saturation on these testbeds. Furthermore, the lack of a large-scale annotated dataset also limits the development of more efficient 3D backbones. Our experiments demonstrated that using a cutting-edge backbone does not automatically translate into improved performance. This surprising trend is even more visible in the synthetic to real-world scenario, for which more research efforts are needed.

## 4.2 Broader Impact

We hope that our research will have a positive impact on both academia and society. In terms of academic research, we emphasized the importance of investigating Open Set scenarios for 3D deep learning. In this context, our benchmark will serve as a reliable starting point for novel methods capable of leveraging the plethora of information naturally offered by 3D data. We release our code with the aim of providing a foothold for future work towards building trustworthy systems that can manage the challenges of the open world. In terms of societal impact, we anticipate that increased academic interest in this field will drive the development of more robust models for safety-critical applications where 3D sensing could be a valuable ally: autonomous driving, robotics and health care are only some examples.

# References

[57] J. Sun, Q. Zhang, B. Kailkhura, Z. Yu, C. Xiao, and Z. M. Mao. Benchmarking robustness of 3d point cloud recognition against common corruptions. *arXiv preprint arXiv:2201.12296*, 2022.