# OpenReview forum: "3DOS: Towards 3D Open Set Learning - Benchmarking and Understanding Semantic Novelty Detection on Point Clouds"
_NeurIPS.cc/2022/Track/Datasets_and_Benchmarks — NeurIPS 2022 Datasets and Benchmarks _

### Official Review · Reviewer_qpmC · 2022-07-21
**Review: Towards Open Set 3D Learning: Benchmarking and Understanding Semantic Novelty Detection on Pointclouds**

**Rating:** 6
**Confidence:** 4
**Correctness:**
**Clarity:** The paper is well written and easy to…

**Strengths:**

- The paper tackles an important problem that has been largely ignored by the research community so far, namely the detection of OOD samples in point cloud classification. The provided benchmark takes a good first step into filling this gap.

- The authors evaluate numerous existing methods that are well-selected. Their benchmark indicates that existing OOD methods from the 2D domain do not transfer well to 3D. This leaves room for future research to improve in this area.

**Weaknesses:**

- My main concern is that all methods are trained on exclusively synthetic samples. The benchmark lacks an evaluation of methods on point cloud data obtained from real sensors. Furthermore, all samples are of relatively low resolution (1024 3D points). Hence, an evaluation on this benchmark leaves the question how well the results transfer to more realistic real-world settings.

- The synthetic benchmark consists of three subsets that are described as increasingly difficult in the paper (l.152). However, I did not find an explanation as to why one subset should be more difficult than another?

**Additional Feedback:**

- The FPR95 values for almost all evaluated methods seem extremely high. In the Synthetic-to-Real benchmark, the top performing methods report a false positive rate of around 70% to 80%. Is there an explanation for why the performance is this poor?

**Documentation:**



**Ethics:**

I do not have any concerns.

**Relation To Prior Work:**

Prior work is sufficiently described.

**Summary And Contributions:**

The paper presents a benchmark for out-of-distribution detection on 3D point clouds. Their benchmark comprises two tracks. The first one is purely synthetic and based on the ShapeNetCore dataset. In the second one, the training set is also synthetic and based on ModelNet40, while the test set contains real point clouds from ScanObjectNN. The authors benchmark multiple OOD methods and reveal room for future improvement.

---

> ### Author Response · Authors · 2022-08-10
> **Response**
>
> **[W1]** Collecting large real-world 3D datasets is costly, indeed up to our knowledge ScanObjectNN is the only real-world 3D dataset that covers a sizable amount of object categories and that was previously used for 3D object classification.
> The point clouds of this dataset contain 2048 points, and in general, the cardinality of point clouds used for 3D object classification is 1024 [30]. With respect to scene datasets this resolution might appear low but it is actually the standard for 3D object classification.
>
> To complete our study we have included a Real to Real Benchmark: we point the reviewer to the initial general answer above that presents and discusses the corresponding results.
>
>
> **[W2]** The task difficulty in the OOD detection context mainly depends on the semantic and visual similarity between known classes and unknown ones. Existing works in this field hardly mention the problem of task complexity or they simply distinguish among two cases [R1]:
> - far OOD: known samples and unknown ones are very different and the task is easy;
> - near OOD: known samples and unknown ones are similar and the task is difficult.
>
> Quantifying this similarity and, as a result, the task difficulty is not a trivial problem. For this reason, most of the papers that use the near/far OOD notation do not specify any numerical measure they used to assign these names.
> In order to solve this problem, it is necessary to define a similarity measure through which to compare known and unknown classes.\
> An effort in this direction has been made by Winkens et al. in [R2] where they proposed the Confusion Log Probability (CLP) metric. CLP is based on the likelihood that an outlier will be confused with an inlier by a classifier trained on both OOD and ID data.
> The rationale behind this choice is that a classifier trained on all the classes will learn features perfectly suited to represent both known and unknown classes. As a result, this oracle classifier should be able to provide a similarity measure between the two set of classes.\
> Unfortunately, we found that training a model on all classes (known+unknown) is also the main weakness of the CLP metric: the representation learned by considering all of the classes is too different from the one that is learned on known classes only in the downstream experiments.\
> As a result, the CLP metric is not really informative about the unknown detection task complexity. We can verify this by computing its value for all of the shifts of our Synthetic to Synthetic Benchmark and comparing it with the baseline AUROC.
>
> We consider here the two most significant values:
>
>
> | Shift      | CLP    | MSP AUROC |
> |------------|--------|-----------|
> | SN1 -> SN3 | -5.970 | 74.6      |
> | SN3 -> SN2 | -1.77  | 94.1      |
>
> Low CLP values indicate low similarity between OOD and ID data (the separation should be easier), whereas higher CLP values indicate higher similarity between OOD and ID data (the separation should be harder).\
> In our experiments, we can clearly see that a shift with a low CLP, which should identify an easy task, is associated with a lower AUROC value when compared to another shift with a higher CLP.
>
>
> An alternative metric with respect to CLP could be obtained by exploiting a natural language model to estimate the semantic similarities among classes. This kind of solution is used in [44] to create “hard”, “medium” and “easy” splits by exploiting the labeled attributes of their chosen datasets.
> However, in our case we can rely only on class names. We performed this experiment using the Natural Language Toolkit [R3] to estimate the similarity between all possible pairs of class names. By computing the average similarity between unknown class names and known ones it is possible to obtain an estimate of the task difficulty. Unfortunately, even this solution does not provide a good measure:
>
>
> | Shift      | Natural language similarity | MSP AUROC |
> |------------|-----------------------------|-----------|
> | SN1 -> SN3 | 2.16                        | 74.6      |
> | SN3 -> SN2 | 2.26                        | 94.1      |
>
> Once again the shift which should be the most complex (higher similarity between known and unknown classes) is the one with the highest AUROC value and therefore the less difficult between the two considered.\
> These experimental results highlight the inadequacy of these metrics as measures of the difficulty of the OOD detection task. As a result, we argue that the only fair way to estimate task difficulty is to look directly at the AUROC trend.
>
>
> [R1] Fort et al., “Exploring the Limits of Out-of-Distribution Detection”, NeurIPS 2021
>
> [R2] Winkens et al., “Contrastive Training for Improved Out-of-Distribution Detection”, Arxiv, 2020
>
> [R3] Bird et al., “Natural language processing with Python: analyzing text with the natural language toolkit.”, O’Reilly Media, Inc., 2009

---

> ### Author Response · Authors · 2022-08-10
> **Response Continued**
>
> **[About FPR95]** The AUROC and FPR95 metrics are computed by using the predicted normality score for all test samples, sorting them and then comparing them against ground truth binary known/unknown labels.\
> However, there is a big relevant difference in how these sorted scores are used in the two cases, so that when testing a given model it is possible to get a good AUROC value with a corresponding bad FPR95.\
> The AUROC value depends on the ordering of the whole set of the normality scores and it is a threshold-free metric: the AUROC can be seen as the probability that a known test sample has a greater normality score than an unknown one.\
> Differently, the FPR95 metric requires computing a known-unknown separation threshold, which is the highest possible value for which 95% of known samples are predicted as known. Once this threshold has been computed, only a subset of all the test normality scores is selected: the one containing the scores above the threshold. Out of them, the FPR95 indicates the fraction of unknown samples that have been predicted as known (false positives).
>
>
> If there is even a single known class with a number of test samples corresponding to at least 5% of all the known samples, and which is particularly difficult to separate from unknown data, the threshold corresponding to FPR95 will be very low.\
> As a consequence, it is highly likely that there will be unknown samples with a score higher than the threshold (i.e. annotated as known), resulting in a high FPR95. And this can happen even if all the other known samples are correctly separated from the unknown ones (high AUROC).
>
>
>
> If the appearance of the test instances is significantly different with respect to the training data of the same classes it might be particularly challenging to identify them as known.\
> This is exactly what happens in our Synthetic to Real Benchmark experiments, where more than 5% of all known test samples are misclassified as outliers and, predictably, result in a high FPR95.\
> On the other hand, in the Real to Real Benchmark where train and test are drawn from the same data distribution, the FPR95 performance improves.

---

### Official Review · Reviewer_w2dt · 2022-07-25
**This paper presents a new benchmark of OOD detection for 3D point cloud.**

**Rating:** 6
**Confidence:** 3
**Clarity:** The paper is written clearly.

**Strengths:**

* The paper presents the point cloud open set learning benchmark firstly and considers two benchmark tracks for synthetic and real point clouds.
* The problem definition is clear, the metrics for evaluating the performance are fairly correct and datasets are sufficient for point cloud OOD detection.
* A thorough analysis of 2D OOD detection methods in 3D situation is explicitly demonstrated in experiments. The results show that 3D open set learning has space for improvement, since SOTA 2D methods are not good enough in  3D situation.

**Weaknesses:**

1) This paper builds a new benchmark for 3D open set learning, which is necessary for 3D literature. However, there is little discussion about the difference between the 2D and 3D open set learning. One obvious difference is the input data structure. Besides, the benchmark almost depends on the 2D counterpart. I think the discussion about the challenges for building 3D compared to 2D open set learning is essential.
2) The performance metrics need more detailed definitions with formulations and citations, since the paper build a new benchmark for a new task.
3) The baseline approach MSP [18] also uses the AUPR metric for evaluation and the mean Pred. Prob as a comparison for the advantages of new metrics (AUROC, AUPR). But this paper only use the AUROC. I think authors need an explanation for metric selection, since a reasonable benchmark should include comprehensive metrics.
4) For the synthetic to real benchmark, it introduces the domain shift in the task. However, the benchmark is for evaluating OOD detection and focusing on discriminating OOD examples. The domain gap is another variable influencing the results in Tab. 3. A real to real track may better analyze the 3D open set learning for comparison.

**Additional Feedback:**

I recommend a borderline-accept for this paper due to the weaknesses. Therefore, the judgement is subject to change according to the other reviews and the rebuttal.

**Correctness:**

The evaluation methods and experiments design are correct but not enough, as mentioned in the weakness 4.

**Documentation:**

Yes, the authors provide codes for reproducing the results.

**Ethics:**

No ethical concerns.

**Relation To Prior Work:**

The differences from 2D open set learning need more discussion.

**Summary And Contributions:**

This paper focus on building a benchmark on OOD detection task for point cloud. It is the first benchmark for open set 3D learning. The proposed benchmark considers two tracks: synthetic -> synthetic and synthetic -> real, which is reasonable for the 3D scenario. To better analyze the situation on OOD detection, authors adopt series of 2D OOD detection methods and directly apply them on point cloud classification methods.  The experiments discover the SOTA method for open set 3D learning, and analyze the advantages and limitations of existing methods.

---

> ### Author Response · Authors · 2022-08-10
> **Response**
>
> **[W1]** We agree with the Reviewer that the difference between 2D and 3D Open Set learning is a crucial point that deserves more space in the paper. Besides the data structure and the need for dedicated feature encoders, the problem of recognizing unknown object categories is much more challenging when dealing with point clouds than with RGB images.\
> **First of all, point clouds often describe only the object geometry, they miss color and texture which are very informative cues in images.** In most of the cases the point cloud datasets contain synthetic objects in isolation: this means that the relative dimension of an object, as well as its context, are missing.\
> In the revised version of the paper, Fig 1 shows the CAD models and relative point clouds of a dishwasher and of a microwave from ShapeNetCore, next to the images of the same objects from ImageNet.
> By focusing only on the point clouds it is difficult to understand whether they are the same object or not, while the differences are much more visible among the images. Note also that the point cardinality has a similar role to image resolution but a small reduction of the point cardinality may lead to a much more significant loss of details than what would happen on images.\
> We discussed these aspects at the beginning of section 3 in the revised paper.
>
> **[W2]** We agree that the description of the AUROC and FPR95 metrics could be more exhaustive. We have included the following in section 3.1 of the revised paper:
>
> Given that the detection of unknown samples is a binary task, both metrics are based on the concepts of True Positive (TP), False Positive (FP), True Negative (TN), and False Negative (FN).
> * The AUROC  is the Area Under the Receiver Operating Characteristic Curve. The ROC curve is a graph showing the TP rate (TPR) and the FP rate (FPR) plotted against each other [18] when varying the normality threshold. As a result, the AUROC is a threshold-free metric, and it can be interpreted as the probability that a known test sample has a greater normality score than an unknown one.
> * The FPR95 is the FP Rate at TP Rate 95%, sometimes referred as FPR@TPRx with x=95%. This metric is based on the choice of a normality threshold so that 95% of positive samples are predicted as positives (TPR=TP/TP+FN). Then the false positive rate (FPR=FP/FP+TN) is computed using this threshold.
>
>
> **[W3]** There are a bunch of different metrics that can be used to evaluate a model’s ability to detect unknown samples. Besides AUROC and FPR95, one could certainly also use AUPR or even other solutions such as TNR@TPRx. However, including too many evaluation criteria may make numeric comparison difficult, so we preferred to choose only two of them.
> We selected AUROC and FPR95 as they are the ones adopted most often in recent works [21,40] with some even using AUROC only [31,42,44,55].\
> For completeness, we included the AUPR results in Table 1 of the supplementary material showing that
> this metric is in line with the others already presented in the main paper. To guarantee maximum flexibility for further experimental evaluations on the proposed testbed, our GitHub project page includes the code to calculate also the AUPR metric.
>
>
> **[W4]** We thank the Reviewer for the suggestion and we point to the initial general answer above that presents and discusses the Real to Real Benchmark results.

---

### Official Review · Reviewer_sMFe · 2022-07-27
**Propose a open set 3D learning benchmark with extensive evaluations.**

**Rating:** 5
**Confidence:** 3
**Correctness:** Yes, the experiment design are approp…
**Clarity:** Yes, the paper is easy to follow. Tho…

**Strengths:**

1. This paper proposes a first testbed for open set 3d learning, which will facilitate the future research and applications on robots and autonomous systems;
2. This paper investigates several algorithms extensively and revealing their strengths and limitations for further research on open set 3d models;

**Weaknesses:**

1. The PointNet++ backbone always outperforms DGCNN on the synthetic to real benchmark while DGCNN outperforms the PointNet++ on the synthetic benchmark, could the authors provide more deep analysis on it;
2. The synth → real benchmark seems to be very sensitive to data augmentation tricks, I doubt whether the dataset settings are proper and consistent to verify the capability of the models.
3. The overall novelty is limited since it only re-organizes three existing 3D datasets and conducts experiments with several existing methods;
4. Results on real → real benchmark is desired;
5. Some typos should be fixed, e.g., L115: Y_t should be Y^t, L328: Tab. 4.3 should be Tab. 4.

**Additional Feedback:**

No additional feedbacks.

**Documentation:**

Yes.

**Ethics:**

Yes.

**Relation To Prior Work:**

Yes.

**Summary And Contributions:**

The authors introduce a novel open set 3d learning testbed based on 3 existing 3D datasets, with several experiment settings in terms of category semantic shift and also covers both in-domain and cross-domain scenarios. They also provide extensive comparisons on these benchmarks and give detailed analysis over them.

---

> ### Author Response · Authors · 2022-08-10
> **Response**
>
> **[W1]** PointNet++ and DGCNN aggregate local information in different ways. The first exploits geometric point neighborhoods (i.e. [x,y,z] distances), while the second considers also per-point feature similarity, thus performing the aggregation in the representation space.
> As a consequence, DGCNN captures relations among points that may be geometrically far away from each other, but semantically close (e.g. chair armrests).\
> At the same time, the feature encoding of DGCNN is more sensitive to noise than that of PointNet++, which becomes evident in the cross-domain (Synthetic to Real) experiments.
> Similar behavior has also been highlighted in a concurrent work [R1], which claims that PointNet++'s robustness to noisy inputs is due to the fixed radius ball query neighborhood aggregation adopted.\
> Furthermore, we invite the Reviewer to check the Real to Real results discussed in the initial general answers which confirm this trend.
>
>
> **[W2]** Unfortunately, we are not sure to understand the point raised by the Reviewer.\
> By their nature Real-World 3D data significantly differs from Synthetic data, thus we adopted data augmentation as a basic strategy to improve model generalization and possibly alleviate the domain shift.
> The augmentation approach makes the training data closer to the test samples by introducing some noise and occlusions. Indeed the obtained results showed that data augmentation has a beneficial effect on the DGCNN results (see the previous answer about the known sensitivity of DGCNN to noise), leading to an improvement of its MSP (+7.2 AUROC points) and SubArcFace (+3.5 AUROC points) performance. On the other hand, the PointNet++ backbone which is much more robust to noises presents almost unchanged results for MSP and shows only a slight decrease for SubArcFace (-1.9 AUROC points).\
> Overall we do not see any inconsistencies that could justify doubts about the capability of the models. Still, to avoid confusion we prefer to move these data augmentation results in the supplementary material, leaving room for the more interesting Real to Real Benchmark experiments in the main paper.
>
>
> **[W3]** With our work, we aim at introducing the open set task in the 3D learning literature. The problem of distinguishing among multiple known categories while identifying unknown ones has been largely overlooked on 3D data despite being crucial for models that operate in real-world conditions, and it is particularly relevant for safety-critical applications.
> Towards this goal, we find it meaningful and appropriate to reorganize and repurpose existing datasets: in this way we defined a reliable benchmark testbed and an experimental protocol through which several open set approaches can be fairly compared.
>
> We highlight that
>
> - our work is in line with the call for papers of the  NeurIPS “Datasets and Benchmarks Track” that welcomes systematic analyses on new problems even without the introduction of completely new datasets.
>
> - although we used the expression ‘existing methods’, it refers to OOD and Open Set approaches developed for 2D data. Moving some of them to work on 3D may be straightforward once the input data structure is properly handled by a point cloud feature encoder, while others (e.g. all the ones including generative strategies ARPL+CS, NF,  OE+mixup) need ad-hoc solutions that we designed for our study.
> We also remark that the representation method SubArcFace belongs to the face retrieval literature and was evaluated before neither for 2D nor for 3D Open Set problems.
>
> - the GitHub project page contains the data and code used for our experiments. It allows the readers to reproduce our results and use them as starting points for new tailored methods with the possibility of assessing them via several performance metrics on large testbeds.
>
> **[W4]** We extended our analysis and presented the Real to Real Benchmark as discussed in the initial general answer.
>
> **[W5]** We thank the reviewer for having pointed out the typos. We updated the text accordingly.
>
>
>
> [R1] Sun et al., “Benchmarking robustness of 3d point cloud recognition against common corruptions”, arXiv preprint, 2022

---

### Official Review · Reviewer_aApU · 2022-07-28
**Great motivation and supporting experiments**

**Rating:** 7
**Confidence:** 4
**Correctness:** No problems at all.
**Clarity:** Well written. (Mentioned on the Stren…

**Strengths:**

- [S1] Although I’m not an expert in this area, I did not have problems of reading this article. The paper is well-written and claims are followed by supporting evidence.
- [S2] Well-designed experiments and tasks. Motivation is clear and paper is well organized.
- [S3] They experimentally demonstrate that traditional methods on Open Set 2D task do not fit on Open Set 3D task, indicating there are many challenges that researchers should solve in the proposed task.

**Weaknesses:**

- [W1] Changing the number of points can also affect the performance. In other words,  both train and test set share the number of points on data so that we cannot clearly say that the splited sets have different domains. Considering real-world situation such as autonomous driving, the authors should also handle this issue.
- [W2] According to Table 1, the reported score does not show clear tendency. For me, it was hard to interpret the message from the Table 1. It would be better to change the table to graph and put the table on the supplementary materials.
- [W3] For Section 3.3, many contents here are repeated from Section 2.2. It would be better to put the details on the supplementary materials.

**Additional Feedback:**

Well-orgnaized paper.
It would be better to add experiments regarding the number of points.

**Documentation:**

Yes.

**Ethics:**

Yes.

**Relation To Prior Work:**

Yes.

**Summary And Contributions:**

- [C1] Open Set Learning, which is a task of classificing data over the known categories but that are OOD, is widely studied in 2D. In contrast, there were no 3D dataset or benchmarks that attempt to solve the Open Set problems.
- [C2] They forumalize Open Set Learning problem and benchmark typical methods.  They evaluate two scores: AUROC(Area Under the Receiver Operating Characteristic) and FPR95(False Positive Rate of OOD examples when choosing a normality threshold for which the true positive rate of in-distribution example is 95%).
- [C3] They organize their data from weel known 3D objects datasets: ShapeNerCore, ModelNet40, and ScanObjectNN. On top of the proposed data, they also organize two benchmarks: synthetic to synthetic(S2S) and synthetic to real(S2R) benchmark.
- [C4] They evaluate various approahces that are feasible to handle challenges of Open Set Learning. They compare for 1) discriminative models, 2) density and reconstruction based models, 3) outlite exposure with OOD generated data, and 4) representation and distance based models.

---

> ### Author Response · Authors · 2022-08-10
> **Response**
>
> **[W1]** We agree with the reviewer that the number of points in train and test data can differ and that this is one of the primary causes of domain shift for 3D data.\
> Our Synthetic to Real Benchmark covers this case. ModelNet40 (train) is composed by 1024-dim point clouds sampled from synthetic CAD models. Instead, the real-world (test) ScanObjectNN samples are composed of 2048-points point clouds. For the real-world dataset the object meshes are not available and the data samples come directly in the form of point clouds, each with 2048 points.\
> We also highlight that, while the synthetic ModelNet40 point clouds exhibit a clean geometry of the foreground object, the same is not true for the real-world ScanObjectNN point clouds. Out of the 2048 points of each test sample, we do not know a priori which are belonging to the foreground object (the labeled one), background or other interacting objects. To ease the replication of our results and to avoid the randomness introduced by any point sub-sampling, we originally decided to test on all available test points (2048), despite the training being performed on lower-resolution (1024 points) synthetic point clouds.
> We have clarified this difference in point cardinality of the considered datasets at the beginning of section 4.1 of the revised paper.
>
> For a more comprehensive analysis, we also ran the Synthetic to Real Benchmark experiments with random-sampled subsets of 512 and 1024 points for the test point clouds. We executed three runs for each experiment and we report the average AUROC results below.
>
> |              | DGCNN 1024-2048* | PN2 1024-2048* | DGCNN 1024-1024 | PN2 1024-1024 | DGCNN 1024-512 | PN2 1024-512 |
> |--------------|:----------------:|:--------------:|:---------------:|:-------------:|:--------------:|:------------:|
> | MSP          |       66.7       |      75.6      |       70.2      |      76.5     |      58.5      |     74.3     |
> | MLS          |       65.7       |      74.8      |       70.4      |      76.4     |      61.9      |     75.2     |
> | ODIN         |       65.7       |      76.0      |       70.1      |      78.3     |      57.4      |     76.1     |
> | Energy       |       65.6       |      74.8      |       70.6      |      77.6     |      62.5      |     76.3     |
> | GradNorm     |       63.4       |      73.0      |       70.5      |      76.5     |      62.2      |     75.5     |
> | ReAct        |       65.6       |      74.6      |       67.5      |      74.3     |      60.8      |     73.1     |
> | Cosine proto |       57.9       |    **78.2**    |       70.0      |      76.2     |      61.1      |     74.9     |
> | CE (L2)      |       66.1       |      77.7      |     **73.9**    |      77.0     |    **64.9**    |     72.4     |
> | SubArcface   |     **71.6**     |      76.9      |       61.9      |    **78.5**   |      55.0      |   **76.8**   |
>
> (\* are results depicted in Tab. 3 of the revised paper; each column represents the AUROC score for backbone *num train points*-*num test points*)
>
> We can conclude that forcing train and test data to have the same number of points (1024-1024) slightly reduces the domain shift and provides a small performance improvement with respect to the remaining cases that present an asymmetry in the point cloud cardinality (1024-2048, 1024-512). These results have been included in the updated version of the supplementary material: see Table 3 and section 2.4. All the novel parts of the supplementary material are in blue font.
>
> **[W2]** We agree with the reviewer that having a visual summary of the results can be helpful. For this purpose, we introduce the plots in Fig. 6 that present the DGCNN and PointNet++ AUROC performance of the baselines and of the most promising methods for all three benchmarks.\
> Besides noticing that PointNet++ is more reliable than DGCNN when dealing with real-world data, we also observe that for some of the approaches the gap between the results obtained with the two backbones is more evident than for others. To identify the best method in each benchmark, we focused on the one presenting the highest mean AUROC performance over the two backbones (see the dashed black line with black star marks).
>
> **[W3]** We agree that sections 2 (we do not have section 2.2) and 3.3 had some overlap. We believe that the related work section (2) should be inclusive enough to provide an overview of the existing literature on OOD detection and Open Set learning, so we maintained it in its original form. We instead adjusted section 3.3 which is now more compact and keeps the most relevant information about the methods that have been selected for the experiments. Both sections are important for a reliable benchmark paper so that the reader can get a comprehensive picture of the state-of-the-art and gather useful insights about the implemented strategies.

---

### Author Response · Authors · 2022-08-10
**To all reviewers**

We thank the reviewers for their constructive feedback. They acknowledged that the paper tackles for the first time the important problem of Open Set 3D learning by introducing a new testbed and presenting an extensive experimental analysis.

In the following, we answer to some general doubts, as well as to all the specific concerns of the Reviewers, and we point to the updated version of the paper that contains new figures and tables. All the updates in the revised version of the paper appear in blue font.

Collecting large real-world 3D datasets is costly, indeed up to our knowledge ScanObjectNN is the only real-world 3D dataset that covers a sizable amount of object categories and that was previously used for 3D object classification.  To overcome this issue, several studies are currently investigating the potentialities and limits of models trained on Synthetic data when deployed on Real-World data. In our work, we followed the same direction by presenting the Synthetic (to Synthetic) and Synthetic to Real benchmarks.

**Still, the Reviewers sMFe, w2dt, and qpmC asked about an analysis of models trained and tested on real-world data.**  To address this request we designed the Real to Real Benchmark. We ran experiments by using the SR1, SR2, and SR3 sets from ScanObjectNN, considering, in turn, two of them as known categories and the third as unknown.

The results are presented in the revised version of the paper in Table 4 and discussed in section 4.4. Overall the trend of the results is similar to what was observed in the Synthetic to Real case. The Cosine proto approach which already showed to be effective with PointNet++ in the Synthetic to Real benchmark, now presents the top results both for DGCNN and PointNet++. We also highlight that PointNet++ maintains a better performance than DGCNN confirming its robustness when dealing with noisy and corrupted real-world data (see also the answer to point (1) of reviewer sMFe).

The different behaviour of the two considered backbones is also evident in the AUROC plots presented in Figure 6 in the revised version of the paper. **We created these plots to provide a summary overview of all three experimental settings**.

By comparing the top results we can rank the Synthetic to Real Benchmark as the most challenging case, followed by the Real to Real Benchmark and finally by the Synthetic Benchmark. This confirms that the domain shift between synthetic and real data adds extra challenges over the semantic shift. Moreover, it is interesting to notice that the improvement provided by the best open set methods over the MLS/MSP baselines is quite visible in the Synthetic Benchmark (DGCNN SubArc > MLS, +3.1 AUROC), but reduces in the Synthetic to Real (PointNet++ CosP > MSP, + 2.6 AUROC) and Real to Real cases (PointNet++ CosP > MLS, + 0.6 AUROC), which clearly asks for new tailored approaches and reveals room for future improvements.

We have added the instructions to get the Real to Real benchmark on the GitHub page of the project to allow full reproducibility of the results.

---

### Author Response · Authors · 2022-08-26
**To all reviewers (2)**

Dear reviewers, with our responses we have addressed your comments, also including further experiments, analyses and visualizations.\
We have just uploaded a new version of the supplementary material which we carefully revised following your feedbacks. Few small corrections have been added also to the revised version of the main paper with a final upload.\
We thank you once again for your useful feedbacks and look forward to discussing any other concern you may have.

---

### Meta-Review · Area_Chair_U5Fy · 2022-09-08

**Recommendation:** Accept
**Confidence:** 5

**Metareview:**

This work for the first time proposes to study the task of Open Set 3D Learning for 3D point cloud data. The authors have conducted extensive experiments under different settings with varied category semantic shifts and provided comprehensive experiments benchmarking the popular methods from 2D open set learning which leads to some important conclusions about the transferability of the 2D methods to the 3D settings. The contribution of the work is clear and novel. The AC believes the paper provides important findings for the community to be aware of.

During the rebuttal, reviewers raised up concerns regarding the lack of real-to-real setting and the missing of some evaluations/metrics. The authors have added the requested materials in the revised paper, which addressed most of the raised issues. While there are some minor questions asked by the reviewers, the authors have carefully addressed them and the AC does not think they are major issues preventing me from accepting this paper. The final scores are 3 accepts and 1 reject, and two reviewers confirmed their final decisions after rebuttal. The reviewer who gave the reject review didn't come back for responding to the authors' rebuttal and the AC think the authors have addressed his/her questions well.

Therefore, the AC is very confident in recommending an acceptance of this work to the track. But, please carefully revise the final paper for the camera-ready submission, based on the reviewers' suggestions. Congratulations!

---

### Decision · Program_Chairs · 2022-09-16

Accept